# Toward the Intelligent, Safe Exploration of a Biomimetic Underwater Robot: Modeling, Planning, and Control

**DOI:** 10.3390/biomimetics9030126

**Published:** 2024-02-21

**Authors:** Yu Wang, Jian Wang, Lianyi Yu, Shihan Kong, Junzhi Yu

**Affiliations:** 1Department of Automation, Tsinghua University, Beijing 100084, China; yu_wang@tsinghua.edu.cn; 2The Laboratory of Cognitive and Decision Intelligence for Complex System, Institute of Automation, Chinese Academy of Sciences, Beijing 100190, China; jianwang@ia.ac.cn (J.W.); yulianyi2022@ia.ac.cn (L.Y.); 3The School of Artificial Intelligence, University of Chinese Academy of Sciences, Beijing 100049, China; 4The State Key Laboratory for Turbulence and Complex Systems, Department of Advanced Manufacturing and Robotics, College of Engineering, Peking University, Beijing 100871, China; kongshihan@pku.edu.cn

**Keywords:** biomimetic underwater robot, obstacle avoidance, deep reinforcement learning, backstepping

## Abstract

Safe, underwater exploration in the ocean is a challenging task due to the complex environment, which often contains areas with dense coral reefs, uneven terrain, or many obstacles. To address this issue, an intelligent underwater exploration framework of a biomimetic robot is proposed in this paper, including an obstacle avoidance model, motion planner, and yaw controller. Firstly, with the aid of the onboard distance sensors in robotic fish, the obstacle detection model is established. On this basis, two types of obstacles, i.e., rectangular and circular, are considered, followed by the obstacle collision model’s construction. Secondly, a deep reinforcement learning method is adopted to plan the plane motion, and the performances of different training setups are investigated. Thirdly, a backstepping method is applied to derive the yaw control law, in which a sigmoid function-based transition method is employed to smooth the planning output. Finally, a series of simulations are carried out to verify the effectiveness of the proposed method. The obtained results indicate that the biomimetic robot can not only achieve intelligent motion planning but also accomplish yaw control with obstacle avoidance, offering a valuable solution for underwater operation in the ocean.

## 1. Introduction

The marine environment constitutes a pivotal realm on Earth, endowed with copious natural resources, including mineral deposits and biological assets of significance. Due to these valuable resources, it is commonly believed that the 21st century is the century of the oceans, marking a period where humanity engages in large-scale exploration and utilization of marine resources. With the deepening development of marine resource exploitation, tasks such as underwater resource exploration and subsea target search in complex aquatic environments are placing increasing demands on the precision of autonomous motion. Therefore, there is an urgent need to advance key technologies in system design, autonomous motion planning, and control methods for underwater robots, ensuring their effective adaptation to the dynamically evolving conditions prevalent in the underwater environment [1,2].

Traditional underwater robots mostly rely on propeller propulsion, benefiting from its advantages of high thrust and speed. However, they often encounter challenges like large disturbances, high destructiveness, and poor maneuverability, preventing them from seamlessly integrating with the natural environment. In recent years, with the development of biomimetics and robotics, significant progress has been made in underwater biomimetic robots [3,4,5]. Drawing inspiration from different marine organisms, various biomimetic platforms have emerged, such as robotic tuna [6], robotic sharks [7], robotic manta [8], robotic dolphins [9], and so on [10]. Each type of underwater biomimetic robot exhibits characteristics such as high maneuverability, low disturbance, and environmentally friendly adaptability. Therefore, utilizing underwater biomimetic robots for autonomous exploration tasks in complex environments holds promise as a paradigm for the future of underwater robotics.

With the improvement of robot computational performance, intelligent navigation and obstacle avoidance algorithms are developing rapidly. Existing methods mainly regard the above problem as a multi-objective optimization process. For the underwater vehicle path-planning problem, Han et al. designed a comprehensive coverage path-planning obstacle-avoidance algorithm aimed at achieving complete coverage of entire sea areas [11]. Based on the genetic algorithm, Wen et al. presented a fusion of heuristic algorithms aimed at accelerating convergence and enhancing feasibility and flexibility [12]. By incorporating autonomous underwater vehicles (AUVs) into the A-star algorithm, Zhang et al. considered dynamic constraints with position information to ensure safety and feasibility [13]. The aforementioned approaches demonstrate global search capabilities; however, they exhibit poor adaptability to the environment and limited local planning capabilities. In recent years, learning-based methods have garnered attention due to their robust nonlinear approximation capabilities. Chen et al. established a dynamic neural network model and further designed a path-planning algorithm that adapted to strong currents [14]. He et al. introduced the asynchronous multithreading proximal policy optimization-based path-planning method, and a goal-distance heuristic reward function was employed to enhance exploration directionality [15]. Yang et al. provided a deep reinforcement learning-based path-planning algorithm, which included a positive experience screening mechanism to improve the reuse rate and dynamic stability [16]. Chu et al. utilized an improved convolutional neural network to construct a deep reinforcement learning method, thus adapting to different dimensional environments [17]. Compared with traditional AUVs, biomimetic underwater robots feature underdriven and strongly coupled motion characteristics. By fully considering their motion constraints, such as maximum steering angular velocity, turning radius, etc., it is conducive to further guaranteeing the safety and reliability of the planning process.

As a key technique for achieving precise tracking of predefined safe paths, tracking control has attracted considerable attention and is primarily divided into model-free control and model-based control. The former has the advantages of simple implementation and fewer requirements for the system model. Cao et al. derived a line-of-sight guidance law and achieved plane path tracking by adjusting the joint phase offset of a robotic snake using a proportional–integral–derivative (PID) motion controller [18]. In conjunction with a sine gait pattern and a proportional–derivative (PD) heading controller, Kelasidi et al. achieved steady-state tracking for a robotic snake in the presence of constant irrotational currents of unknown direction and magnitude [19]. Yu et al. designed a fuzzy PID (FPID) heading controller for straight-line tracking by an underactuated AUV under different velocity profiles [20]. Zhang et al. proposed a motion controller for a Dactylopteridae-inspired biomimetic underwater vehicle with a fuzzy adaptive PID controller, achieving high- and low-speed stable line cruising [21]. However, model-free tracking controllers often exhibit suboptimal transient control performance in some complex environments. Therefore, model-based methods have gradually been applied. Li et al. presented an adaptive path-tracking controller for a multi-joint snake robot based on an improved serpentine curve, leading to fast convergence speed and high stability of the position error [22]. With full consideration of the stochastic disturbances and uncertainties in hydrodynamic parameters, Mahapatra et al. derived an $H_{\inf}$ tracking controller for AUVs to achieve precise tracking of planar polyline paths [23]. Regarding the issue of unknown external disturbances and control input saturation in underwater vehicles, Li et al. developed a robust tracking controller to guarantee convergence within a specified time and a transient tracking error within a predefined boundary [24]. Yan et al. provided a two-dimensional trajectory tracking method for a biomimetic underwater robot subjected to external disturbances, employing robust nonlinear model predictive control [25]. He et al. proposed a robust non-smooth controller to achieve simultaneous tracking of the reference yaw angle and waypoints, which were validated through simulations for docking tasks and bridge pier detection tasks [26]. By constructing a dead-zone compensator and a two-layer cascaded tracking controller, Yu et al. addressed the issue of strong oscillations in torque input and motion tracking of underwater vehicles caused by the inherent dead zone in the propeller thrusters [27].

In this paper, we focus on safe exploration tasks in complex underwater environments, aiming to provide an intelligent motion planning and control framework for biomimetic underwater robots. The main contributions of this paper can be summarized as follows:Aiming to achieve underwater operation in complex environments, an intelligent safe exploration framework is proposed for a biomimetic underwater robot, including obstacle avoidance modeling, motion planning, and yaw control.Taking into account rectangular and circular obstacles, the sensor detection and obstacle collision models are established, providing accurate information for the planner.With regard to intelligent motion planning, a Deep Deterministic Policy Gradient (DDPG)-based planner is provided with the designed state space, action space, and reward function. In particular, the performances of different training setups are investigated. Furthermore, a backstepping-based yaw control law is derived by combining it with a sigmoid function-based smoothing method. Extensive simulations are conducted, demonstrating the effectiveness of the proposed methods.

The remainder of this paper is organized as follows. Section 2 provides a simple explanation of the problem statement and the control framework. Section 3 details the methodologies of the model, motion planner, and yaw controller design. Furthermore, the simulation results are presented, followed by a comprehensive analysis. Finally, Section 5 concludes this paper and discusses future work.

## 2. Problem Statement and Control Framework

Safety obstacle avoidance technology plays a crucial role in autonomous underwater exploration. It can not only prevent collisions with obstacles but also optimize path planning to ensure the efficient arrival of underwater robots in the target area. By integrating advanced sensing technologies, a safety obstacle avoidance system can acquire data on the underwater environment in real time. By performing environmental perception and obstacle detection, timely and accurate decisions can be made to ensure the safety of task execution. Therefore, this paper proposes an autonomous exploration method for biomimetic underwater robots.

Utilizing a biomimetic robotic fish as a vehicle, we conduct research on autonomous exploration methods. Figure 1 illustrates the conceptual design of the biomimetic robotic fish [28]. The safety obstacle avoidance system consists of three ranging sensors oriented toward the front, left, and right directions. In the air, these sensors can acquire distance information within a maximum measurable range of 30 m. However, in water, due to the influence of optical attenuation, the effective measurement range is reduced to 1–2 m. Additionally, the biomimetic robotic fish features a multi-joint tail propulsion system. By oscillating the tail, forward thrust and yawing moment can be generated, thereby accomplishing autonomous underwater locomotion. Further, the designed conceptual computational modules of the robotic fish can be divided into the top decision level and bottom control level. Specifically, the top-level decision-making module, which is applied for deep reinforcement learning planning, employs an NVIDIA Xavier NX edge computing platform, providing high-performance CPU, GPU, and AI computing capabilities for such applications as robots. With regard to the bottom level, an STM32F407VGT6 microcontroller is employed, featuring an ARM Cortex-M4 core capable of meeting various requirements such as high computational performance and low power consumption.

Furthermore, Figure 2 illustrates a schematic diagram of an underwater autonomous exploration task. The biomimetic robotic fish departs from the starting point and moves toward the target point. φ indicates the current yaw angle. $\varphi_{g}$ represents the angle between the current heading and the target heading. $d_{g}$ denotes the distance to the target point. Regarding obstacle measurements, the measuring distance of the three ranging sensors is set to $l_{rader}$, which is smaller than the maximum underwater measurement distance of the utilized sensing devices. Additionally, the installation angles for the left and right ranging sensors are denoted as $\varphi_{install}$. Regarding obstacles, two obstacle types are designed in this paper: rectangular obstacles and circular obstacles. Both types of obstacles can be set at random positions and sizes. Hence, overlapping is permitted to form underwater obstacles of any shape.

Additionally, in order to establish the underwater environment, we employ the simplified kinematic models for the underwater robot, as follows:(1)$$\begin{array}{c} \dot{x}=u\cos\varphi -v\sin\varphi \\ \dot{y}=u\sin\varphi +v\cos\varphi \\ \dot{\varphi }=r \end{array}$$
where $p\left(t\right)=\left(x\left(t\right),y\left(t\right)\right)$ represents the position with respect to the inertia frame. (u,v,r) denotes the linear and angular velocities with respect to the body frame.

Considering that this paper focuses on navigation control tasks in narrow environments, the biomimetic robotic fish in such conditions exhibits relatively low speed but high maneuverability. Therefore, this paper assumes a longitudinal velocity of 0, with a given forward velocity of u= 0.3 m/s. In summary, the dynamic expression in the yaw direction can be formulated as follows:(2)$$\begin{array}{c} \dot{r}=\frac{1}{m_{33}}\left(\tau_{r}+\left(m_{11}-m_{22}\right)uv-d_{33}r\right) \end{array}$$
where $\left(m_{11},m_{22},m_{33}\right)$ and $d_{33}$ are the mass and damping parameters larger than zero. $\tau_{r}$ indicates the yaw moment.

Based on the aforementioned problem description, this paper proposes an autonomous intelligent exploration framework for biomimetic robotic fish, taking obstacle avoidance into account, as shown in Figure 3. Firstly, the underwater environment is modeled. Adopting a sensor design scheme based on biomimetic robotic fish, two types of obstacles, rectangular and circular, are designed, further establishing the obstacle models and collision models. This enables real-time interaction between the biomimetic robotic fish and environmental information. Secondly, using the DDPG method, a neural network based on multi-layer perceptron (MLP) is constructed to achieve intelligent and safe underwater motion planning. Considering the characteristics of the task, the state space, action space, and reward function are designed, and the performances of different training setups are investigated. Thirdly, based on the obtained motion planning, a heading motion control method based on the backstepping approach is provided. Specifically, to address sudden changes in planning quantities, we adopt a smooth transition method based on a smooth step function to enhance tracking stability. In this process, due to the higher real-time requirements of yaw control compared to motion planning, different time periods are applied to the planning layer and control layer.

## 3. The Methodology of Intelligent Safe Exploration

### 3.1. Obstacle Collision Model

Real-time interaction with the underwater environment is a crucial aspect of robot motion planning and control, especially in underwater environments with complex obstacles. This paper introduces two types of obstacles in the environment: rectangular obstacles and circular obstacles. For rectangular obstacles, the defining variables include the center position, length, width, and rotation angle. Similarly, each circular obstacle is characterized by the center position and radius. By calculating the distance between the biomimetic robotic fish and the obstacles, the state inputs are provided for deep reinforcement learning. Next, we offer a detailed explanation of the collision model.

Firstly, the detection model for three ranging sensors is established, as illustrated in Figure 4. According to the aforementioned description, the ranging sensors are oriented in different directions with varying installation angles. The choice of this angle directly influences the range of detection. Therefore, a well-designed installation angle contributes to maximizing the efficiency of the safety system, ensuring stable and comprehensive distance data provision in various scenarios. Furthermore, since the utilized ranging sensors belong to single-line lidar, their detection models can be described by geometric line segments. The starting point of the line-segment models for these three sensors is the real-time position of the biomimetic robotic fish, and the detection distance is denoted as $l_{rader}$. Consequently, the end positions of these line segments can be derived as follows:(3)$$\begin{array}{c} P_{s}=\left(x+{\tilde{l}}_{rader}*\cos\left(\varphi \right),y+{\tilde{l}}_{rader}*\sin\left(\varphi \right)\right)\\ P_{l}=\left(x+l_{rader}*\cos\left(\varphi +\varphi_{install}\right),y+l_{rader}*\sin\left(\varphi +\varphi_{install}\right)\right)\\ P_{r}=\left(x+l_{rader}*\cos\left(\varphi -\varphi_{install}\right),y+l_{rader}*\sin\left(\varphi -\varphi_{install}\right)\right) \end{array},$$
where ${\tilde{l}}_{rader}=l_{rader}\cos\left(\varphi_{install}\right)$. The purpose of this configuration is to ensure that the endpoints of the three ranging sensors align in a straight line. Since the forward sensor’s detection direction corresponds to the swimming direction of the robotic fish, if the detection distance is set the same as that of the sensors on both sides, the effectiveness of the side sensors for detecting obstacles in the forward direction may be limited. Consider a scenario where the biomimetic robotic fish traverses the bottom-right corner of a rectangular obstacle. When the only forward sensor obtains collision information, the robot’s movement direction is determined by the current heading angle, potentially resulting in a left or right turn. However, a leftward movement may result in ongoing collisions with the obstacle. Conversely, by aligning the endpoints of the three sensors’ detection points, simultaneous detection of the obstacle can be ensured by both the left and forward sensors. In such a scenario, the robot opts to move right to circumvent the obstacle.

Furthermore, for the three ranging sensors, the line-segment detection model can be expressed as follows:(4)$$P_{o}P_{s,l,r}\to \left\{x,y,l_{rader},\varphi ,\varphi_{install}\right\}.$$

Next, a brief description of the modeling process for rectangular obstacles is presented. In Figure 4, the center point of rectangular obstacle *i* is denoted as $O_{rec}^{i}=\left(x_{orec}^{i},y_{orec}^{i}\right)$, with the length and width represented by $len_{orec}^{i}$ and $wid_{orec}^{i}$, respectively. Consequently, the coordinates of the four corners of rectangular obstacle *i* can be derived as follows:(5)$$\begin{array}{c} A^{i}=\left(x_{orec\text{-}A}^{i},y_{orec\text{-}A}^{i}\right)=\left(x_{orec}^{i}-\frac{len_{orec}^{i}}{2},y_{orec}^{i}+\frac{wid_{orec}^{i}}{2}\right)\\ B^{i}=\left(x_{orec\text{-}B}^{i},y_{orec\text{-}B}^{i}\right)=\left(x_{orec}^{i}-\frac{len_{orec}^{i}}{2},y_{orec}^{i}-\frac{wid_{orec}^{i}}{2}\right)\\ C^{i}=\left(x_{orec\text{-}C}^{i},y_{orec\text{-}C}^{i}\right)=\left(x_{orec}^{i}+\frac{len_{orec}^{i}}{2},y_{orec}^{i}-\frac{wid_{orec}^{i}}{2}\right)\\ D^{i}=\left(x_{orec\text{-}D}^{i},y_{orec\text{-}D}^{i}\right)=\left(x_{orec}^{i}+\frac{len_{orec}^{i}}{2},y_{orec}^{i}+\frac{wid_{orec}^{i}}{2}\right) \end{array}.$$

Further, considering rectangular obstacles with different attitudes, four corner points can be rotated. Taking corner point *A* as an example, the coordinate position of the rotated point can be derived as follows:(6)$$A^{\prime }=\left(\begin{array}{c} {x_{orec\text{-}A}^{i}}^{\prime }\\ {y_{orec\text{-}A}^{i}}^{\prime } \end{array}\right)=\left(\begin{array}{c} \left(x_{orec\text{-}A}^{i}-x_{orec}^{i}\right)\cos\left(\alpha \right)-\left(y_{orec\text{-}A}^{i}-y_{orec}^{i}\right)\sin\left(\alpha \right)+x_{orec}^{i}\\ \left(x_{orec\text{-}A}^{i}-x_{orec}^{i}\right)\sin\left(\alpha \right)+\left(y_{orec\text{-}A}^{i}-y_{orec}^{i}\right)\cos\left(\alpha \right)+y_{orec}^{i} \end{array}\right),$$
where α is the rotation angle. Therefore, based on the aforementioned description, the four sides of the rectangle can be modeled. Each side can be represented by a triplet, including the starting point of the line segment, the length of the line segment, and the angle of the line segment, as follows:(7)$$\begin{array}{c} AB\to \left\{A,wid_{orec}^{i},-90{}^{\circ}+\alpha \right\}\\ AD\to \left\{A,len_{orec}^{i},\alpha \right\}\\ CB\to \left\{C,len_{orec}^{i},-180{}^{\circ}+\alpha \right\}\\ CD\to \left\{C,wid_{orec}^{i},90{}^{\circ}+\alpha \right\} \end{array}.$$

The circular obstacle can be described by its algebraic equation as follows:(8)$${\left(x-x_{ocir}^{i}\right)}^{2}+{\left(y-y_{ocir}^{i}\right)}^{2}={r_{ocir}^{i}}^{2}.$$

Finally, collision detection modeling is conducted based on the distance sensor detection model and the obstacle models. Algorithm 1 provides the detailed process for collision modeling. The input comprises the detection model and obstacle model, and the output consists of real-time feedback distance information from three distance sensors. Specifically, for rectangular obstacles, the algorithm calculates the intersection points between the line-segment models of the three distance sensors and the edge line-segment model of the rectangular obstacle. This process yields the distances between the robot and each intersection point. Next, by sorting the distances, the minimum value is chosen as the input for the sensor data. The same methodology is applied to circular obstacles.
**Algorithm 1** Obstacle collision model**Input:** $\left(x_{orec}^{i},y_{orec}^{i}\right)$, $len_{orec}^{i}$, $wid_{orec}^{i}$, α, $\left(x_{ocir}^{i},y_{ocir}^{i}\right)$, $r_{ocir}^{i}$**Output:** The distances measured by the three distance sensors to the nearest obstacles $d_{s}$, $d_{l}$, $d_{r}$.1:Obtain the line-segment model of the range sensors and the line-segment model of each obstacle $\left\{P_{o}P_{s}\right\}$, $\left\{P_{o}P_{l}\right\}$, $\left\{P_{o}P_{r}\right\}$.2:For rectangular obstacles *i*, the set of boundary models is $\left\{A^{i}B^{i},A^{i}D^{i},C^{i}B^{i},C^{i}D^{i}\right\}$, whose slope and intercept are $k_{j}^{i}$ and $c_{j}^{i}$, j=AB,AD,CB,CD.3:Determine whether $\left\{P_{o}P_{s}\right\}$ and $\left\{A^{i}B^{i},A^{i}D^{i},C^{i}B^{i},C^{i}D^{i}\right\}$ are parallel.4:**if** parallel **then**5:    It indicates no intersection points.6:**else**7:    Calculate the intersection point $\left(x_{c},y_{c}\right)=\left(\frac{c_{j}^{i}-c_{l}}{k_{l}-k_{j}^{i}},\frac{k_{l}\left(c_{j}^{i}-c_{l}\right)}{k_{l}-k_{j}^{i}}+c_{l}\right)$, and obtain the distance $d_{l}$.8:**end if**9:Sequentially verify whether the intersection point lies on the two intersecting line segments simultaneously.10:**if** True **then**11:    Output $d_{l}$.12:**else**13:    It indicates safety.14:**end if**15:For circular obstacle *i*,16:**if** 
$∥P_{l}O_{cir}∥>r_{ocir}^{i}$ **then**17:    It indicates safety.18:**else**19:    Compute the intersection points between the line and the circle and determine the one closer to the robot as the obstacle intersection point. Next, obtain the distance $d_{l}$.20:**end if**21:Verify whether the intersection point lies on the line segment.22:**if** True **then**23:    Output $d_{l}$.24:**else**25:    It indicates safety.26:**end if**27:For the forward and right sensors, use the same steps as the left sensor.

### 3.2. DDPG-Based Motion Planner

Based on the above obstacle collision model, we conduct motion planning in this section. Taking into account the safety, smoothness, and shortest path of the motion, we design the DDPG algorithm to achieve autonomous obstacle avoidance motion in a multi-obstacle environment. Firstly, the autonomous obstacle avoidance motion can be regarded as a Markov decision process, as well as a multi-objective optimization problem. In recent years, deep reinforcement learning algorithms have gained widespread attention for their powerful nonlinear optimization capabilities and have gradually been applied to robot control tasks. The DDPG algorithm has demonstrated its superiority in handling high-dimensional, continuous action spaces. In detail, the actor network is responsible for outputting actions, whereas the critic network evaluates the value of these actions. Through the collaborative work of the actor and critic, DDPG can more effectively learn policies in complex environments, achieving optimization in continuous action spaces. Therefore, this paper adopts the DDPG method to conduct motion planning, allowing the robot to learn appropriate actions in a multi-obstacle environment. Next, based on the above algorithm, the training framework for a fully connected neural network using MLP is constructed by designing the multi-variable state space, action space, and reward function. By judiciously selecting the number of layers and nodes of the fully connected neural network, a balance between the expressive power of the model and computational efficiency can be achieved. Additionally, we construct the multi-layer perceptron considering a lightweight network architecture. There are just two hidden layers, in which the numbers of neural nodes are 400 and 300, respectively.

#### 3.2.1. State Space

The reasonable design of the state space is crucial for network training in reinforcement learning. Based on the defined task, this paper introduces five state variables, as follows:$d_{g}$ represents the Euclidean distance between the current position of the robot and the target point, which aims to guide the robot toward motion planning along the shortest distance. Additionally, the maximum value of this distance is determined by the distance between the starting point and the target point.$\varphi_{g}$ represents the difference between the current heading of the robot and the target heading, which primarily guides the robot to move optimally in the desired direction. Therefore, the range of this value is [−180°,180°].$d_{s}$ represents the real-time obstacle measurement information from the forward sensor.$d_{l}$ represents the real-time obstacle measurement information from the left sensor.$d_{r}$ represents the real-time obstacle measurement information from the right sensor.

Furthermore, we normalize and scale the aforementioned state variables to accommodate different initial and target values.

#### 3.2.2. Action Space

Based on the kinematic model of biomimetic robotic fish, we consider the yaw angle increment Δφ as the output value of the neural network. The determination of the output range is derived from the yaw angular velocity of the biomimetic robotic fish. Given a motion planning period of $T_{p}=0.2$ s, the network output η rad at a specific moment implies that the turning angular velocity should reach $\eta /T_{p}$ rad/s.

In conjunction with our prior works, considering the dual factors of obstacle avoidance and protection of the drive mechanism in narrow environments, the turning angular velocity of the biomimetic robotic shark is approximately 50°/s. Therefore, we set the range of the action space as $\Delta \varphi \in \left[-10{}^{\circ},10{}^{\circ}\right]$. Nevertheless, based on the aforementioned rationale, it seems that the yaw angular velocity is also another reasonable network output. There are two reasons for selecting the yaw angle increment rather than the angular velocity as the output. On one hand, the yaw angular velocity is susceptible to disturbances and data instability in practical applications. Utilizing it as the input for subsequent controllers may lead to system instability. Filters are often applied in practical scenarios, introducing new issues related to signal lag. On the other hand, utilizing the yaw angle as the input allows for smoother processing in subsequent controllers, facilitating control law design.

#### 3.2.3. Reward Function

The formulation of the reward function directly influences the strategies learned by the agent, guiding the robot toward optimization across multiple objectives. By incorporating a balanced consideration of reward weights for different objectives, a multi-objective reward function is designed, which can be summarized as follows:$r_{1}=-d_{g}$: The value of the distance variable in the state space is employed as the first reward term, aiming to achieve the characteristic of the shortest path.$r_{2}=-\left|\varphi_{g}\right|$: The difference in the target heading is utilized as the second reward term, with the intention of directing the robot toward the target heading.$r_{3}=-\left|{\varphi_{g}}^{\prime }\right|$: The derivative of the difference in the target heading is employed as the third reward term, facilitating the smoothing of the heading output angle, thereby obtaining a smooth trajectory.$r_{4}=-\left(\frac{1}{d_{s}}+\frac{1}{d_{l}}+\frac{1}{d_{r}}\right)$: The summation of the reciprocals of the real-time distances between the robot and the nearest obstacle, measured by three sensors, is employed as the fourth reward term. The aim is to encourage the robot to maintain a certain distance from obstacles as much as possible during its motion.$r_{5}=f\left(x,y\right)$: This term represents the terminal reward, providing a relatively large positive reward when the robot reaches the goal point. The definition is as follows:
(9)$$f\left(x,y\right)=\left\{\begin{array}{c} 1\;\;\;if\sqrt{{\left(x-x_{g}\right)}^{2}+{\left(y-y_{g}\right)}^{2}}<0.1\\ 0\;\;\;otherwise \end{array}\right..$$

By linearly summing the aforementioned reward terms with appropriate weights, the overall designed reward can be expressed as follows:(10)$$R=\sum_{i}^{5}\kappa_{i}r_{i},$$
where $\kappa_{i}$ denotes the weight coefficients, which are also normalized.

Based on the previous setup, a deep reinforcement learning framework can be developed for network training. To expedite training, a training environment based on line-segment obstacles is employed. Considering the single-line distance measurement characteristic of the sensors, specific point information can be collected. Hence, obstacles are represented as line segments rather than actual rectangles or circles during network training. This design offers two advantages. Firstly, adopting the form of line-segment obstacles can reduce training complexity to some extent, thereby enhancing training speed. In comparison with the utilization of real rectangular or circular obstacles, the geometric simplicity of line-segment obstacles streamlines the training process, rendering it more efficient. Secondly, line-segment obstacles can be set at arbitrary angles, allowing for substantial intersection among these segments. Consequently, with a sufficient number of line segments, the theoretical complexity of the obstacle environment surpasses that of regular shapes (rectangles or circles). To a certain extent, this setup facilitates the reinforcement learning network in acquiring obstacle avoidance strategies more accurately in practical environments, thereby improving its generalization performance in real-world scenarios.

Building upon this foundation, we delve into various training environment configurations, with a primary focus on whether the position of obstacles changes during a single training iteration. Specifically, two training strategies are explored. Firstly, a strategy is employed where the positions of obstacles remain fixed throughout the entire training process. Theoretically, this strategy is anticipated to yield relatively stable training outcomes due to the uniqueness of the optimal path. Secondly, a dynamic configuration is introduced, incorporating a triggering variable to initiate changes in obstacle positions. Specifically, the initial obstacle configuration is set, and training commences. Within a single episode, when the robot reaches the target point, the triggering variable increments. Upon the triggering variable surpassing a predefined threshold *M*, the obstacle positions are reconfigured. In this paper, we investigate the training outcomes for different values of *M*, specifically *M* = 1, 20, and 50. Such a design allows for a comprehensive exploration of the impact of diverse training environments on model performance.

### 3.3. Backstepping-Based Yaw Controller

Based on the motion planner, we can obtain the incremental value $\Delta \varphi \left(t\right)$ of the target yaw angle, thus acquiring the target yaw angle $\varphi_{d}\left(t\right)=\varphi_{d}\left(t-1\right)-\Delta \varphi \left(t\right)$ at time *t*. For ease of expression, $\varphi_{d}$ is used hereafter to refer to the target yaw angle. As described in the action space design, the obtained target heading angle may exhibit significant discontinuities, posing higher demands on the controller’s design. Therefore, to achieve smoother changes in the target heading signal, this paper proposes a smoothing method for handling step changes in control targets.

The sigmoid function, characterized by its smooth continuity, is a type of S-shaped function that performs exceptionally well in processing step signals. By adjusting the parameters of the sigmoid function, the smoothness of the step signal can be flexibly controlled to meet the requirements of different control systems. Thus, this paper adopts the sigmoid function as a smoothing term, aiming to reduce the rate of change of the target heading signal by introducing weight coefficients. This smoothing step method contributes to alleviating the system’s abrupt load, enhancing the stability and performance of the controller. The expression of the smoothing process is as follows:(11)$${\hat{\varphi }}_{d}\left(t\right)=\left(\varphi_{d}-{\tilde{\varphi }}_{d}\right)\cdot f_{s}\left(t\right)+{\tilde{\varphi }}_{d},$$
where
(12)$$\begin{array}{c} f_{s}\left(t\right)=\frac{1}{1+e^{-\lambda t}}\\ t\in \left[0,T_{c}\right] \end{array},$$
where ${\tilde{\varphi }}_{d}$ indicates the primeval yaw angle based on the action output of the planner. λ denotes the coefficient.

Furthermore, the yaw controller is designed based on the backstepping method [29]. First, define the yaw error as follows:(13)$$\begin{array}{c} e_{\varphi }=\varphi -{\hat{\varphi }}_{d}\\ {\dot{e}}_{\varphi }=\dot{\varphi }-{\dot{\hat{\varphi }}}_{d}=r-{\dot{\hat{\varphi }}}_{d}=\xi +e_{r}-{\dot{\hat{\varphi }}}_{d} \end{array}.$$

Next, define the Lyapunov function $V_{1}$ for $e_{\varphi }$, and derive its derivative as follows:(14)$$\begin{array}{c} V_{1}=\frac{1}{2}e_{\varphi }^{2}\\ {\dot{V}}_{1}=e_{\varphi }{\dot{e}}_{\varphi }=e_{\varphi }\left(r-{\dot{\hat{\varphi }}}_{d}\right)=e_{\varphi }\left(\xi +e_{r}-{\dot{\hat{\varphi }}}_{d}\right) \end{array}.$$

Further, in order to achieve $e_{\varphi }\to 0$, we can design the following:(15)$$\xi ={\dot{\hat{\varphi }}}_{d}-k_{1}e_{\varphi },$$
where $k_{1}$ represents the coefficient. Hence, the derivative can be formed as follows:(16)$${\dot{V}}_{1}=-k_{1}e_{\varphi }^{2}+e_{\varphi }e_{r}.$$

Then, define the Lyapunov function $V_{2}$ for $e_{r}$, and derive its derivative as follows:(17)$$\begin{array}{c} V_{2}=V_{1}+\frac{1}{2}e_{r}^{2}\\ {\dot{V}}_{2}={\dot{V}}_{1}+e_{r}{\dot{e}}_{r}={\dot{V}}_{1}+e_{r}\left({\dot{e}}_{r}\right)=e_{r}\left(\dot{r}-\dot{\xi }\right) \end{array}.$$

Based on the dynamic model, $\dot{r}$ can be expressed as $\dot{r}=\left(\tau_{r}+\left(m_{11}-m_{22}\right)uv-d_{33}r\right)/m_{33}$. Therefore, we can design the control law as follows:(18)$$\tau_{r}=\left(m_{22}-m_{11}\right)uv+d_{33}r+m_{33}\left(\dot{\xi }-k_{2}e_{r}-e_{\varphi }\right).$$

Consequently, we can obtain the following:(19)$${\dot{V}}_{2}=-k_{1}e_{\varphi }^{2}-k_{2}e_{r}^{2}.$$

Obviously, $\forall e_{\varphi }\neq 0,e_{r}\neq 0$, $V_{2}<0$, indicating that the yaw control system is asymptotically stable.

## 4. Simulation and Analysis

In this section, we carry out extensive simulations to validate the effectiveness of the proposed planner and controller. Firstly, a simulated pool environment is established, along with the network training and testing environment. Further, a multi-layer perceptron based on a fully connected network is constructed with two hidden layers. Additionally, two distinct time intervals are employed in this paper: the planning period $T_{p}$ and the control period $T_{c}$. The key parameters of the planning and control systems can be seen in Table 1. The other parameters, such as the model and training setup of the network parameters, can be found in our previous work [30].

### 4.1. Results and Analysis of the Planner

Figure 5 presents the training results under a fixed obstacle configuration, including a reward and two types of losses. The shaded region in the figure represents the standard error of the data annotated based on the training results. The results indicate that the proposed deep reinforcement learning planning method achieved rapid convergence. Although there is some instability in the first 500 steps, the subsequent curves demonstrate relatively stable performance.

Furthermore, we investigated the relationship between the partial state variables and output actions. Taking $d_{g}$ and $\varphi_{g}$ as examples, Figure 6 indicates the mapping relationship between them. Since the forward velocity was set to a fixed value and the lateral velocity was neglected, theoretically, there was a stronger correlation between the output actions and yaw angle. The training results demonstrate that in terms of the overall trend, the output actions generally changed with the variation of the yaw angles. Additionally, $d_{g}$ also played a role in some aspects. For instance, when $d_{g}$ was relatively large under the same yaw angle error, the output action tended to be more gradual. The main reason might be that when the distance to the target point was considerable, the learned strategy not only pursued the target yaw direction but also considered the smoothness of the path.

In addition, to thoroughly investigate the network’s performance under different training conditions, we conducted tests with different values of *M*. The specific definition of *M* is as follows. During the network training process, if the robotic fish successfully reaches the target point in each episode within the same obstacle environment, it is recorded as one success, and *M* represents the total number of successful attempts. Therefore, by fixing the obstacle environment and setting M= 1, 20, 50, four agents can be obtained. Subsequently, multiple tests were conducted for each training network. By employing the control variable method, we set up eight sets of identical obstacle environments for each agent, enabling the robotic fish to perform motion planning, and recording the cumulative reward. Figure 7 shows the corresponding results, illustrating the maximum and minimum rewards obtained by the four training environments in the eight test environments. Additionally, we also calculated the median, mean, and standard deviation for the four sets of data, as tabulated in Table 2. From the perspectives of the median, minimum, mean, and standard deviation, the network performance obtained from training in a static obstacle environment was optimal. As the value of *M* increased, the performance gradually decreased, which was also validated from the viewpoints of the mean, standard deviation, and minimum values. The plus sign + in figure indicates that the value is automatically marked as an outlier due to the large difference from the median. The reasons for the above phenomenon are mainly twofold. Firstly, when changing the environment, the positions of the obstacles were randomly generated, leading to situations where obstacles were either too simple or complex in certain trials. Secondly, theoretically, the static obstacle environment, equivalent to M→∞, resulted in a relatively unique optimal solution in that environment, facilitating the learning of the optimal strategy. Certainly, due to the total episode number being set at 3000, with a maximum of 300 steps per episode, the limitation on sample quantity may also have influenced the results.

### 4.2. Results and Analysis of Obstacles Avoidance in Complex Environment

In this section, we select the network trained in a fixed obstacle environment for autonomous motion testing. Firstly, by randomly setting obstacle modeling data, a complex obstacle environment was generated. Secondly, the starting point (1.5, 1.5), the target point (6.5, 6.5), and the initial posture of 45° were defined. Subsequently, the DDPG-based planner and backstepping-based controller were applied. The results of the snapshot sequences are shown in Figure 8, indicating that the robotic fish successfully detected and correctly avoided each obstacle during the movement toward the target point. Figure 9 illustrates the real-time variation trends of the five state variables. Specifically, we divided the motion process shown in the figure into three stages based on obstacle avoidance behavior, represented by obstacles of different colors. In the first stage, the biomimetic robotic fish’s left and forward sensors detected obstacles, prompting a right turn based on network output. Since obstacles still existed on the right, the right and forward sensors received information, leading to a left turn. At this point, the robotic fish’s position was between two obstacles, with only the forward sensor having no data, yet the robotic fish chose to move straight ahead. The obstacle avoidance task in the second stage was relatively simple. When the left sensor detected an obstacle, a slight right turn led to successful avoidance. The third stage was similar to the first stage. However, the yaw angle error state variable in Figure 9 exhibited an approximately 5° steady-state error at the initial stage. The main reason might be the presence of a certain sudden change in control torque at the beginning of motion, leading to instability. In the later control stages, despite encountering multiple obstacles, the trained network continued to navigate toward the target.

Additionally, Figure 10 depicts the planning and control output variables during the autonomous exploration process. Regarding the planning output, the results indicated that the trained network tended to adopt relatively strong actions when facing complex narrow environments. However, taking values at the boundaries of the action space may lead to some step changes. Fortunately, the employed smooth transition method alleviated this issue to some extent. As shown in Figure 10, it can be observed that when the planning output underwent a step-like change, the proposed method enabled a smooth transition, thereby improving the stability of the control input to a certain extent. In terms of the control output, it is evident that during the obstacle avoidance stage, there was a noticeable change in the control output, allowing the robot to instantly output a large yaw moment to complete turning actions. In summary, the above process thoroughly demonstrates that the proposed method can successfully achieve autonomous, safe, and intelligent motion planning and control in narrow and multi-obstacle environments.

### 4.3. Discussion

With the aid of high maneuverability, a robotic fish has the ability to move in a complex and narrow environment that is full of obstacles. Hence, an autonomous intelligent planning and control framework for a biomimetic robotic fish is proposed for autonomous exploration tasks in complex multi-obstacle environments. Considering two common types of obstacles, i.e., rectangular and circular, we establish an obstacle collision model with three distance sensors installed on a robotic fish. On this basis, we design a motion planner and yaw controller based on deep reinforcement learning and backstepping methods, successfully achieving intelligent safe exploration.

Despite successfully achieving autonomous obstacle avoidance for intelligent exploration, there are some aspects that still require improvement. Firstly, we regard the biomimetic robotic fish as a point, without considering its shape and size information. Therefore, shape modeling of the biomimetic robotic fish is part of our ongoing work. Secondly, DDPG faces the issue of easily becoming stuck in extreme action regions and easily learns strategies at the edges of the action space. To address this problem, considering a new network architecture or incorporating concepts from other deep reinforcement learning algorithms should be considered.

## 5. Conclusions and Future Work

In this paper, we have developed an intelligent safe exploration framework for biomimetic robotic fish, which includes an obstacle avoidance model, DDPG-based motion planner, and backstepping-based yaw controller. With regard to the obstacle avoidance model, an obstacle detection model was established using three onboard distance sensors. Next, a collision model considering rectangular and circular obstacles was constructed, further obtaining the distances to the nearest obstacle. Utilizing these distances as the input states, a DDPG-based motion planner was designed to output the target yaw motion, with obstacle avoidance considered in the reward design. Using the action output, a sigmoid function-based smooth method was applied to generate the stable target yaw angle. Further, a backstepping-based method was employed to achieve yaw control. Finally, extensive simulations were carried out, demonstrating the effectiveness of the proposed method.

In the future, we will develop a prototype of the robotic fish, including the sensor system, top decision-level module, and bottom control-level module, and conduct aquatic experiments to further verify the effectiveness of the proposed methods. Additionally, we will focus on underwater stereoscopic vision to identify obstacles, thereby obtaining more abundant environmental information. Further, a vision-based navigation algorithm is another area worthy of study.

## Figures and Tables

**Figure 1 biomimetics-09-00126-f001:**
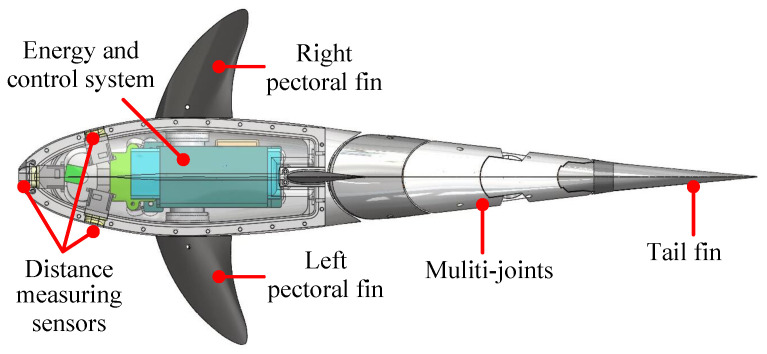
The conceptual design of the biomimetic robotic fish.

**Figure 2 biomimetics-09-00126-f002:**
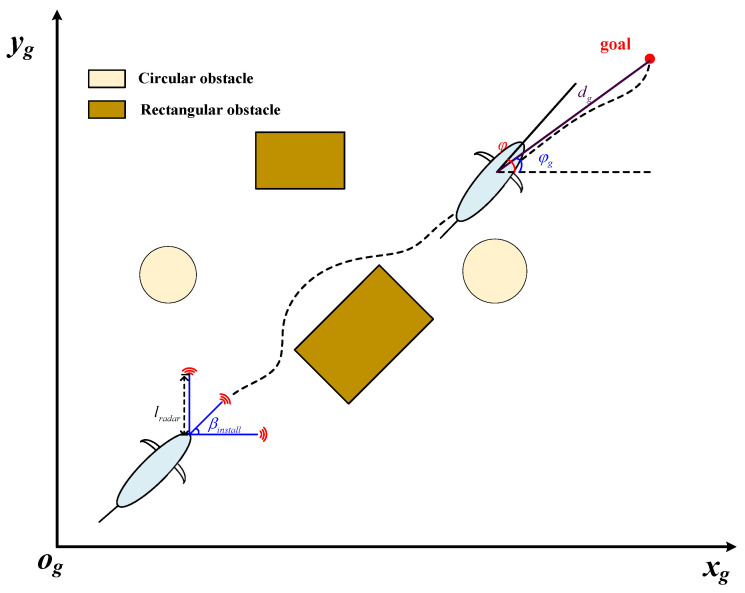
A schematic diagram of an underwater autonomous exploration task.

**Figure 3 biomimetics-09-00126-f003:**
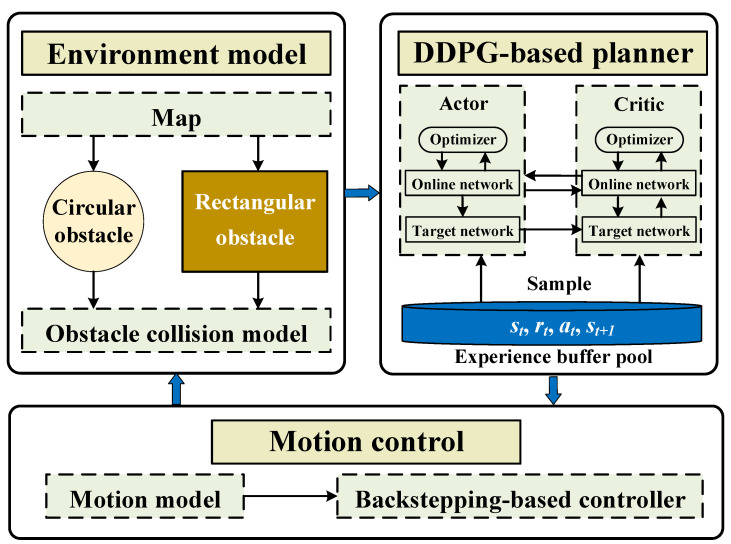
The control framework of autonomous intelligent exploration.

**Figure 4 biomimetics-09-00126-f004:**
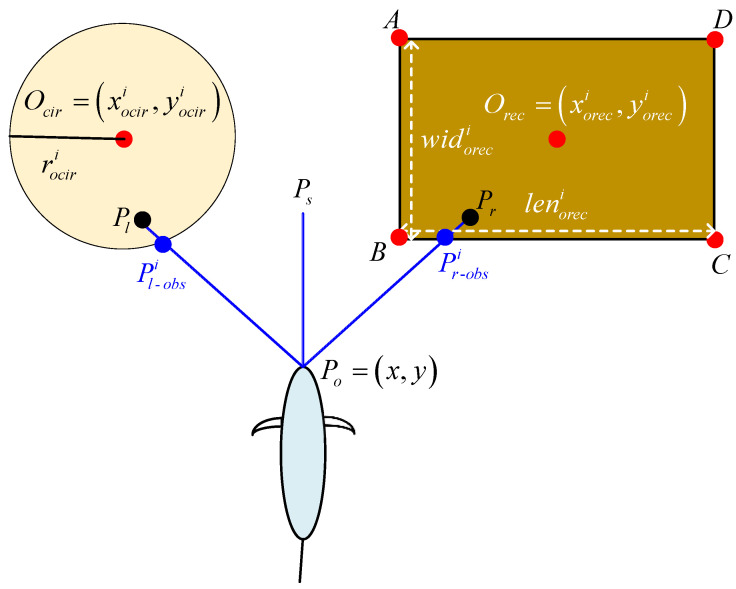
An illustration of the obstacle detection model.

**Figure 5 biomimetics-09-00126-f005:**
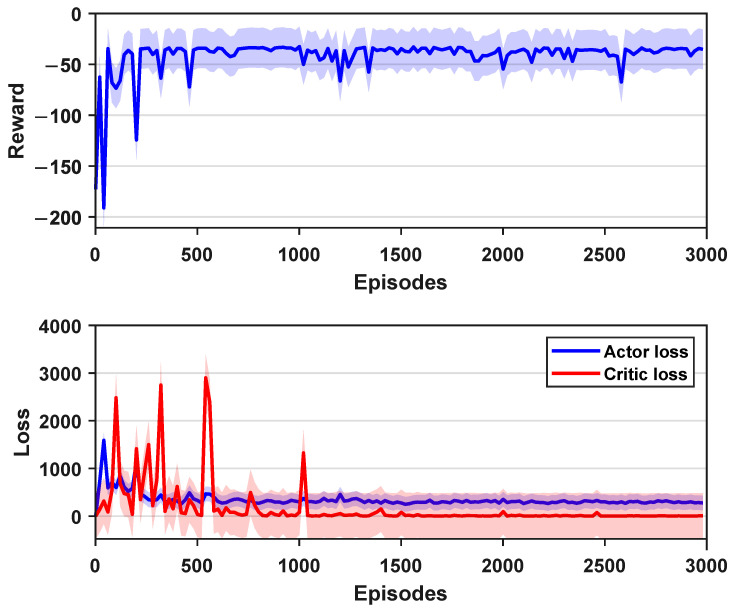
The training results of the DDPG-based planner.

**Figure 6 biomimetics-09-00126-f006:**
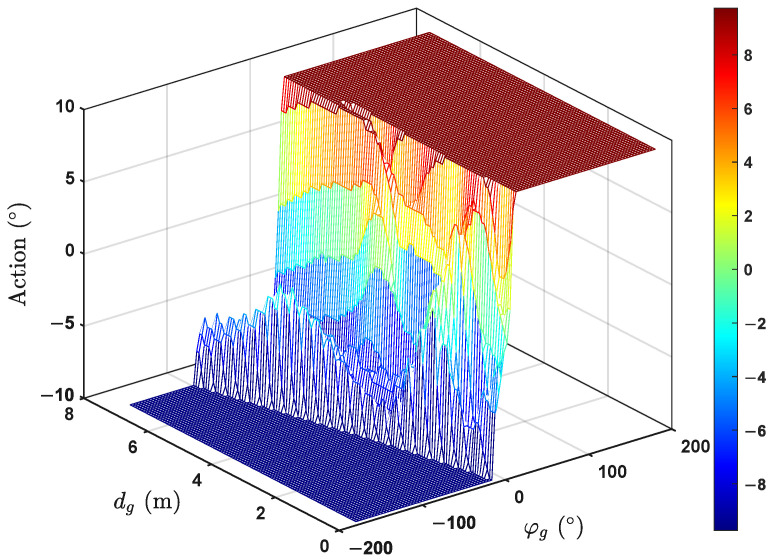
The relationship between the partial state variables and output actions.

**Figure 7 biomimetics-09-00126-f007:**
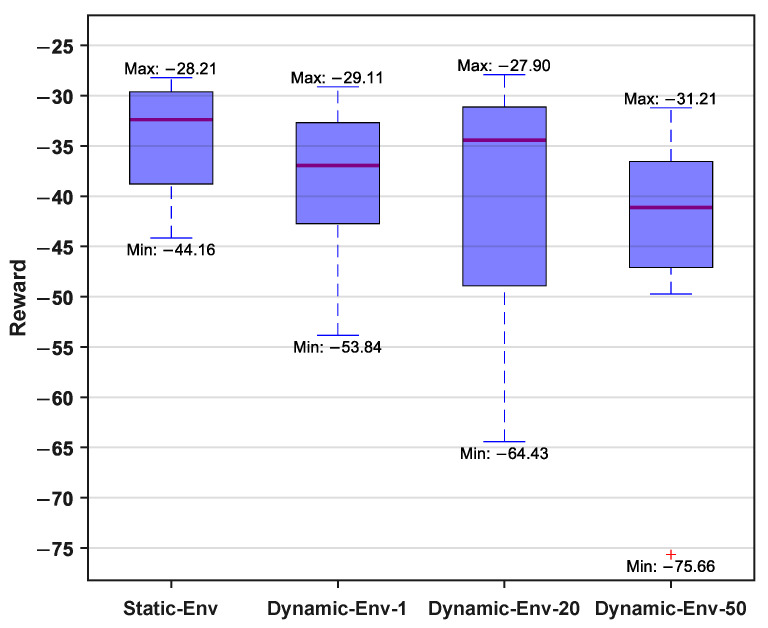
The results of testing different training setups.

**Figure 8 biomimetics-09-00126-f008:**
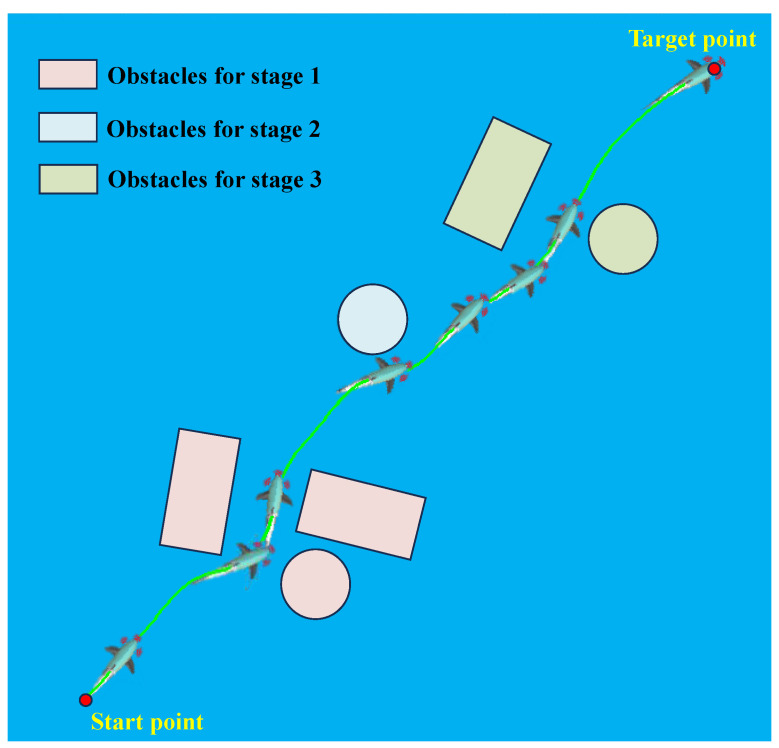
The snapshot sequences of intelligent safe exploration.

**Figure 9 biomimetics-09-00126-f009:**
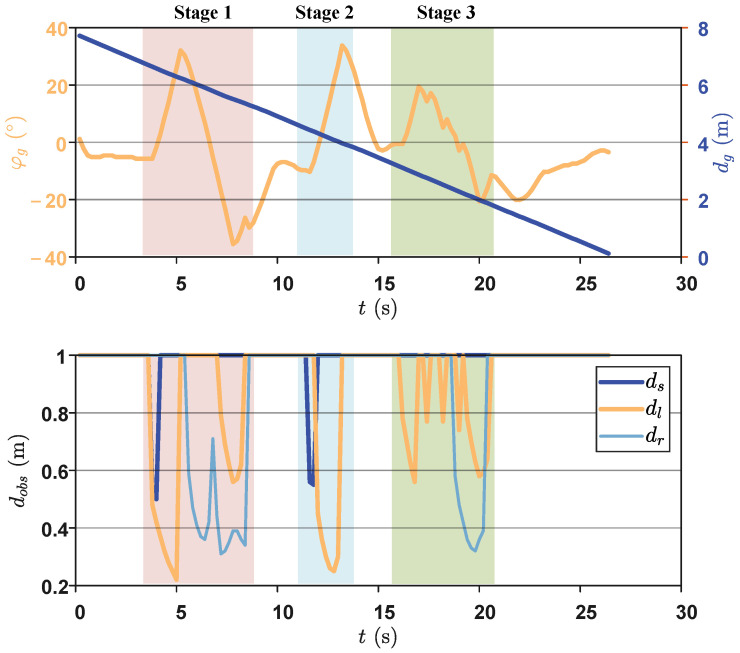
The real-time variation trends of the five state variables.

**Figure 10 biomimetics-09-00126-f010:**
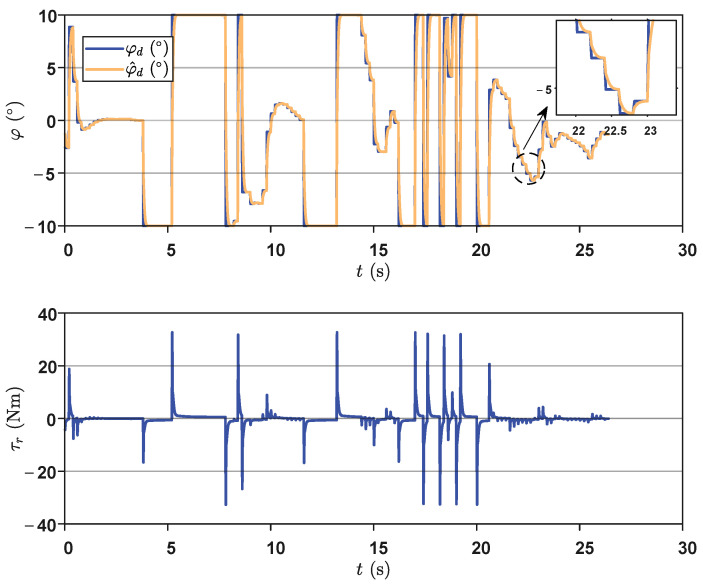
The output results of the planning and control variables.

**Table 1 biomimetics-09-00126-t001:** The parameters of the planner and controller.

**Item**	$\kappa_{1}$	$\kappa_{2}$	$\kappa_{3}$	$\kappa_{4}$	$k_{1}$	$k_{2}$	λ	$T_{c}$	$T_{s}$	$\varphi_{install}$	$l_{rader}$
**Value**	0.4	0.4	0.05	0.15	2	2	30	200 ms	20 ms	45°	0.8 m

**Table 2 biomimetics-09-00126-t002:** Results analysis of different training environments.

Data	Static-Env	Dynamic-Env-1	Dynamic-Env-20	Dynamic-Env-50
Median value	−32.385	−36.945	−34.425	−41.13
Mean value	−34.243	−38.460	−40.161	−44.553
Standard error	5.940	8.125	13.668	13.834

## Data Availability

The data generated during the current study are available from the corresponding author upon reasonable request.

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
