# Peer review of "Toward the Intelligent, Safe Exploration of a Biomimetic Underwater Robot: Modeling, Planning, and Control"

_biomimetics, 2024, doi:10.3390/biomimetics9030126_

Round 1

Reviewer 1 Report

Comments and Suggestions for Authors

The paper is written clearly and concisely. The novelty of the work seems to be okay. Underwater robots are not new but the way authors proposed deep learning for obstacle detection is appreciable. My concern is, that the robot is simulated and works fine but how it will be experimented on? Because it requires high computing resources as DL is incorporated. In addition, the robot has to make decisions dynamically. In such conditions, how the controller will work, i.e. how it handles the situation? The authors need to comment on this aspect. In addition, I feel the literature study needs some more updates. Therefore, the authors are instructed to improve the literature study by referring to newly published papers. Please check for grammatical mistakes and typo errors. The future study must be updated. 

Comments on the Quality of English Language

Needs minor corrections

Author Response

First of all, we greatly appreciate the valuable feedback. We have taken sincere care of the issues raised by you and thoroughly revised the manuscript. We highlighted the revised description in the manuscript. Please check the response to each comment in the attached response file.

Reviewer 2 Report

Comments and Suggestions for Authors

The paper is well presented and technically sound.

Underwater safe exploration in ocean is addressed in this paper. An intelligent underwater exploration framework of a biomimetic robot is proposed including the obstacle avoidance model, motion planner, and yaw controller.

Firstly, with the aid of the onboard distance sensors in robotic fish, the obstacle detection model is established. Secondly, a deep reinforcement learning method is adopted to plan the plane motion, and the performances of different training setup are particularly investigated. Thirdly, a backstepping method is applied to derive the yaw control law, in which a sigmoid function-based transition method is employed to smooth the planning output.

With the considerations of rectangular and circular obstacles, the sensor detection and obstacle collision models are particularly established, providing the accurate information for the planner. With regard to the intelligent motion planning, a Deep Deterministic Policy Gradient (DDPG)-based planner is provided with the designed state space, action space, and reward function.

The future work should focus on the underwater stereoscopic vision to identify obstacles, so as to obtain more abundant environmental information.
Questions posed were addressed and by which specific experiments. The conclusions are consistent with the evidence and the arguments presented.
All main questions posed are addressed by the experiments made.
References are appropriate and recent..

Author Response

(The authors gave the same response as above.)
